# Results of a Qualitative Study Aimed at Building a Programme to Reduce Cardiovascular Risk in People with Severe Mental Illness

**DOI:** 10.3390/ijerph19116847

**Published:** 2022-06-03

**Authors:** Marie Costa, Nicolas Meunier-Beillard, Elise Guillermet, Lucie Cros, Vincent Demassiet, Wendy Hude, Anna Baleige, Jean-François Besnard, Jean-Luc Roelandt, Frédéric Denis

**Affiliations:** 1EPSM Lille-Métropole, Centre Collaborateur de l’OMS Pour la Recherche et la Formation en Santé Mentale, 211 Rue Salengro, 59260 Hellemmes, France; demassiet.vincent@gmail.com (V.D.); hudewendy@gmail.com (W.H.); jean-luc.roelandt@ghtpsy-npdc.fr (J.-L.R.); frederic.denis@univ-tours.fr (F.D.); 2EPSM Lille-Métropole, 104 Rue Général Leclerc, 59280 Armentières, France; 3INSERM, UMR 1123, ECEVE Faculté de Médecine Paris Diderot, Paris 7 Site Villemin, 10 Avenue de Verdun, 75010 Paris, France; 4CHU F. Mitterrand, Délégation à la Recherche Clinique et à l’Innovation, 21000 Dijon, France; nicolas.meunier-beillard@chu-dijon.fr; 5INSERM CIC 1432 Module Epidémiologie Clinique, 21000 Dijon, France; 6Instance Régionale d’Education et Promotion de la Santé, 76100 Rouen, France; e.guillermet@ireps-bfc.org (E.G.); l.cros@ireps-bfc.org (L.C.); 7EA 75-05 Education Ethique Santé, Faculté de Médecine, Université François-Rabelais Tours, 37032 Tours, France; baleige.a@gmail.com; 8Centre Hospitalier de Fougères, 35300 Fougères, France; jfbesnard@ch-fougeres.fr; 9Faculté d’Odontologie, Université de Nantes, 44000 Nantes, France; 10Clinical Research Unit, La Chartreuse Psychiatric Centre, 21033 Dijon, France

**Keywords:** qualitative study, cardiovascular risk, psychiatry, mental health, carers, mental health service users, severe mental illness

## Abstract

People with severe mental illness (PSMI) have a shorter life expectancy and are more likely to have cardiovascular disease than the general population. Patients, carers, psychiatric professionals and primary care providers can all play a role in increasing PSMI physical health. The present qualitative exploratory study aimed to explore the views of these four populations as part of the multi-phase COPsyCAT project, whose objective is to build and test a cardiovascular risk prevention programme for PSMI. Overall, 107 people participated in the study’s 16 focus groups, which were transcribed and analysed in a thematic analysis. With a view to building the health promotion programme, major themes identified in the corpus were translated into a list of needs as follows: communication, information, training and support. Results show that it is essential to improve communication between all the different stakeholders in mental health. The greatest challenge facing this programme will be to adapt it to the needs and expectations of PSMI while facilitating work between the various mental health stakeholders. Simple and inexpensive actions could be taken to improve the cardiovascular health of PSMI and will be experimented with during the programme’s feasibility study which will start in September 2022.

## 1. Background

Severe mental illness (SMI) concerns approximately four million people in France [1]. The life expectancy of people with SMI (PSMI) is 10–25 years shorter than for the general population [2]. The main factors associated with this abnormally high mortality are cardiovascular-related causes [3,4,5]. There is a strong prevalence of behavioural risks for chronic cardiovascular-related disease in PSMI, including poor nutrition, lack of physical activity, tobacco smoking, and alcohol use disorder [6,7,8].

Structural factors also contribute to the high mortality rate in this population, such as access to and the use of mental and physical healthcare services [9], as well as the quality and type of care they receive [10]. Despite the existence of good practice guidelines for care [11], PSMI are still more likely to experience cardiovascular disease irrespective of the type of psychiatric disorder [12]. In this context, the World Health Organisation (WHO) defined the promotion of physical health of PSMI as one of the priorities of the 2013–2020 Global Mental Health Plan [13], and a recent report from the 2020–2030 United Nation’s Sustainable Development Goals agenda [14] was developed with this objective in mind [15].

In the field of mental health, carers play an important role in the provision of care to PSMI [16]. A carer can be defined as an unpaid family member or friend who provides daily help to PSMI in various related areas, including hands-on care, care coordination, and the management of finances [17].

It has been observed that patient and carer empowerment [18] enables better follow-up and better health-related outcomes, including a reduction in cardiovascular disease [18]. There are many definitions of empowerment; a review published in 2016 analysing the various definitions and dimensions of empowerment found that its main dimensions include participation in decision making, gaining control, knowledge acquisition and coping skills [19]. Empowerment can be achieved using a patient/carer-centred approach, and therefore, active participation of both stakeholders is critical [20].

Psychiatric health professionals and primary care providers are also essential stakeholders in caring for PSMI and are strategically placed to promote and implement an empowerment-based approach to care. Furthermore, physical and rehabilitation medicine (PRM) physicians play an important role in interventions and programmes designed to reduce cardiovascular risk [21] and consequently make an important contribution to improving cardiovascular health in PSMI.

Several studies have explored the representations and experiences of patients [22,23], carers [23,24], psychiatric health professionals [25,26] and primary care providers [27] in terms of the physical health of PSMI. However, to the best of our knowledge, no published work to date has focused all four of these populations in the same study. Moreover, we believe that no study to date has explored PSMI empowerment as an approach to use in order to build a comprehensive programme to reduce cardiovascular risk in this population in France.

In this context, we implemented a qualitative, exploratory study conducted on these four populations as part of the multi-phase COPsyCAT study which aims to build and test a cardiovascular risk prevention programme for PSMI. The development of the programme is currently in its final stages.

## 2. Methods

### 2.1. COPsyCAT General Study Design

The present work describes the first component of the larger mixed-methods research project COPsyCAT. Specifically, it describes the prospective multicentre qualitative exploratory phase. The project’s protocol has already been published [28]. The project comprises five steps: (i) creating a steering committee, (ii) developing the qualitative study interview guide, (iii) implementing the qualitative study using focus groups (FG), (iv) constructing a cardiovascular risk prevention programme using results from the qualitative study, and (v) testing this programme in 2022 at the CHU Tours and CH Etampes in France. Patients, carers, psychiatric health professionals and primary care providers are the project’s four target populations and are involved at all study stages, from early design to implementation and data analysis.

### 2.2. Design of the Qualitative Exploratory Study

The qualitative study consisted of sixteen FG with persons from the four target populations. It was conducted in seven mental healthcare facilities in France from June 2019 to April 2021. These institutions are all members of a health cooperation group coordinated by the EPSM–Lille–Métropole–WHO Collaborating Centre for Research and Training in Mental Health.

### 2.3. Participant Recruitment

#### 2.3.1. Patient Recruitment

The principal investigators (all physicians) working at each of COPsyCAT’s five participating sites (one investigator per site) already knew the participants. They identified PMSI who were eligible (i.e., who met inclusion criteria) for enrolment in the qualitative exploratory study and invited them to participate in COPsyCAT during a routine follow-up visit. Those interested in participating were then referred to the somatic physician at the study site who provided them with more detailed information about the study (its aims, potential impact and potential outcomes). The physician also underlined that there would be no impact on their care irrespective of their final decision to participate or not. At the request of a patient, a trusted person could also be present during this meeting. Information concerning data management and the right to withdraw from the study was also provided in the form of a written information note which was handed directly to the patient or to the trusted person. The patient/trusted person could ask any question he/she wished about the study. For this qualitative study, participants were asked to provide oral consent as required by the legal framework for this type of study.

#### 2.3.2. Carer Recruitment

Carers were recruited in the same way as patients. The principal investigator in each site contacted people who met inclusion criteria. Those who expressed their interest received additional information and the information note from the investigator. They subsequently decided to participate or not.

#### 2.3.3. Psychiatric Health Professional and Primary Care Provider Recruitment

Eligible health professionals (psychiatric care and primary care) were identified through hospital reports sent by the participating sites. The investigator at each site contacted them and provided further information to those who agreed to participate.

#### 2.3.4. Inclusion Criteria

For all participants:-Over 18 years old.-Consent to participate.Patients using psychiatric services:-PSMI followed in an inpatient or outpatient setting.Carers-Providing support to a PSMI with a long-term mental condition.-Consent needed from the PSMI receiving support.Psychiatric health professionals:-Psychiatrist or nurse working in a hospital or in a private setting.Primary care providers:-General practitioner, nurse, dietician, or pharmacist following at least one PSMI.

#### 2.3.5. Exclusion Criteria

For all participants:-Inability to participate in an FG for physical or psychological reasons.-Not affiliated with France’s health insurance system.-Under legal protection.-Pregnant, parturient or breastfeeding.-Did not speak or understand French.

### 2.4. Data Collection

Sixteen FG were conducted (four for each target population). The structure of the FG followed the interview guide developed during the first phase of the COPsyCAT study (see above).

Each of the sixteen FG was facilitated by a two-member unit from the research team. NM, a sociologist, was a co-facilitator in all the FG. The other team member was sometimes an anthropologist (MC or LC) and sometimes a peer-researcher (WH). In all FG, the same interview guide was followed according to the study population. FG took place in a room provided by the five healthcare facilities participating in the study. Because of the COVID-19 pandemic and associated preventive measures, some FG were conducted using web video conferencing software. FG were either audio-recorded (face-to-face setting) or video recorded (video conferencing setting).

### 2.5. Analysis

A thematic analysis [29] was performed on the whole corpus using the computer-assisted qualitative data analysis software Nvivo (Melbourne, Australia) by three researchers: two sociologists (LC, NMB) and a public health researcher (MC). As described in Figure 1, the coding process was first performed independently by all three reviewers. Researchers then discussed their coding and the themes and subthemes they identified in each corpus during a triangulation session.

## 3. Results

### 3.1. Description of Participants

A total of 106 persons participated in the 16 FG from June 2019 to April 2021. The average number of participants in each FG was 7 {5; 8}. One of the two researchers in each FG was responsible for facilitating it, while the other kept track of time and ensured the discussions ran smoothly. The average duration of the FG was 80 min.

#### 3.1.1. Users of Psychiatric Services

Of the 26 patients interviewed, 13 (65%) were men, and average age was 48 years. Twenty percent of the patients reported that they had no educational diploma, and 55% declared having an educational level equal to or higher than upper secondary school certificate.

#### 3.1.2. Carers

Eighteen (69%) of the 26 carers interviewed were women, and the average age was 62 years. Eighteen (69%) reported having an educational level equal to or higher than upper secondary school certificate. With regard to their relationship with the PMSI they cared for, 20 (77%) were parents, 1 (4%) was a son or daughter (specific data not available), 3 (11.5%) were partners, and 2 (8%) were siblings.

#### 3.1.3. Psychiatric Professionals

Two (6%) of the 30 psychiatric professionals were general practitioners, and 28 (93%) were psychiatrists. Twenty-one (70%) were female, and the average age was 42 years.

#### 3.1.4. Primary Care Providers

Among the 26 primary care providers, 15 (57%) were women, and the average age was 44 years. One was a psychiatrist (4%), 13 were general practitioners (50%), 2 were dentists (7%), 2 were pharmacists (7%), 4 were nurses (15%) and 4 (15%) were social workers.

### 3.2. Presentation of Themes

As the purpose of the COPsyCAT is to build a cardiovascular risk prevention programme, a ‘needs’-related framework was constructed to classify the following four major common (i.e., for all four target populations) needs which emerged during the FG: (1) communication, (2) information, (3) training and (4) support.

#### 3.2.1. Theme 1—Communication

The need for more communication was the major theme in all four populations’ discourses. This mainly concerned communication between psychiatric professionals and primary care providers. Both populations complained that they did not have access to information about patients, for example, what treatments they were taking and what medical procedures had been performed (blood tests, etc.)


*“Yes, communication, that’s it; what’s missing is communication, but the truth is, it’s up to us to create it.”*
Psychiatric Professional


*Unfortunately, for most patients, we don’t know, we don’t have the diagnoses (…) we have to go and fish a bit for the information. And even if the doctors want to, it’s again a question of the time [needed to send us information]. There are many things that hinder communication between us and them. When we have very important concerns, we phone, but for the most part we now just try to send emails.”*
Primary care provider

These healthcare professionals wanted to see strategies to be developed for the exchange of information about patients, ideally the sharing of the latter’s medical records, but failing that, by email, text message or a liaison booklet.


*“I think that we spend our time looking for information, especially with our patients. Therefore, I think it [shared patient medical record] would save time, a lot of time, because we have to call the secretary.”*
Psychiatric professional


*“We don’t need to know the details. What we’re interested in is the pathologies (…) that the pathology is identified. Really, [having] a shared medical record would be a complete change in the relationship between all the professionals. The aim is for everyone to have access to the minimum amount of information they need.”*
Primary care provider

Carers also expressed many complaints about the low level of communication they received from professionals about their PMSI relative. They also noted a compartmentalisation between psychiatry and general medicine. They wanted interdisciplinary round-table discussions to be set up where health professionals could manage real-world cases.


*“(…) when you have cancer, the doctors hold multidisciplinary consultation meetings. (…) That is to say, no treatment is done without a whole bunch of specialists around the table saying ‘We’re going to do this’ and ‘We’re not going to do that’; they measure, they weigh up, they give the pros and the cons’, etc. I mean, there are enormous resources. (…) But for our relatives, it’s: “I’ll try this for you, oh that’ll make you fat, but well, it’s not serious”.*
Carer

Finally, users of psychiatric services stressed the importance of communication between psychiatrists and primary care providers in order to receive a good medical follow-up.

Pharmacological treatments were an important part of the users’ discourses, with many complaining about side effects. Improving communication was seen as an opportunity to talk about treatment and to implement appropriate changes (i.e., lower doses, treatment changes and treatment interruption).


*“I think that there should be more exchanges between doctors: Between psychiatrists and general practitioners, there should be exchanges, because at the end of the day, when we go to see the general practitioner, he’s not up to speed with everything, he asks us questions.”*
User of psychiatric services

#### 3.2.2. Theme 2—Information

The theme ‘information’ grouped all the information that could be provided without any need for support or feedback (i.e., one-directional communication, for example a website or a brochure). One of the main concerns expressed by patients was the lack of information about the cost of care. Examples of this were information about doctors who collaborate with France’s health insurance system, so that patients pay a subsidized rate, and about doctors who charge extra fees, as health insurance in France does not cover beyond a set rate for a specific type of consultation.


*“So I think I’m going to go back. I was waiting to see if it would be reimbursed, because at that time it wasn’t. But now [it is reimbursed so] that’s good.”*
User of psychiatric services

Carers and primary care providers expressed the need for information on local services and activities suitable for PSMI and their carers (e.g., support groups, sports activities and nutrition information). For carers, this involved finding help to provide support to their loved ones. For providers, it involved orienting their patients and their patients’ carers to places where they could get support.


*“It took me four years to find out that there was a training programme for families and I only found out in October. It was supposed to start in September. It’s true that I waited [before looking for information]. They told me: “No, you have to wait until next year.”*
Carer


*“There is the same thing in B.; they do cooking, eating behaviour, etc. It’s very interesting and it is not expensive (…). They have more and more requests [for this] but it’s from people who already know about it. It’s not the same. It’s not necessarily the people that need it the most who have the information. (…) likewise, we need to be informed, we are not necessarily informed.”*
Primary care provider


*“I don’t have a brochure of the medical-psychological centre that shows me the pathway…. D. tells me that there is a dietician at the medical-psychological centre. I don’t know anything about it actually because… because… yes, I didn’t look for information.”*
Primary care provider

Some of primary care providers who participated also reported that they had too little knowledge of tools such as therapeutic patient education (TPE) and psychoeducation.

Psychiatric health professionals insisted on the need to provide information to primary care professionals about the psychiatric care services their local hospital offered in order to improve their collaboration:


*“(…) We invited all the general practitioners. There’s not a lot of them. We invited them to show them the services we offer, to present what we do, to talk about how you get to the medical-psychological centre, the reality on the ground…”*
Psychiatric professional

Finally, psychiatric health professionals emphasized the importance of relaying France’s national cardiovascular risk screening campaign to patients and of doing this as part of global prevention policy, in order to not to stigmatise them:


*“Without discrimination, (…) without prescription, it needs to be a bit automatic. It must become natural to systematically talk about metabolic risk in the [psychiatric health] services, whether services are open or closed [i.e., patients cannot leave the premises], whether patients have good cognitive efficiency or not, and whatever the severity of their illness.”*
Psychiatric professional


*“(…) If you screen for cardiovascular risks in schizophrenics, maybe you should do like you do for the general population…, I don’t know, we have screening for breast cancer at 50, screening for colon cancer, etc. Perhaps, we should do a more global cardiovascular risk campaign for vulnerable people…”*
Psychiatric professional

#### 3.2.3. Theme 3—Training

Each of the four target populations expressed needs that could be solved with training. Patients explained that they would like to develop skills to be able to talk about their illness. In addition, stigma was mentioned as a barrier to care.


*“It’s difficult. I don’t know how to talk about it [mental disorder]. It would be nice to have more training about that, which is done (…) through talking with other people. I’d like that a lot. It would make me feel better.”*
User

Patients also expressed the wish for more technical knowledge, covering areas such as nutrition and treatment. Treatment appeared to be a major problem for patients. They regretted the fact that they had too little information on, and the opportunity to discuss, how treatments work, dosages and side effects. With regard to treatment, carers and primary care providers mentioned the importance of providing training to both patients and carers.


*Primary care providers expressed a lack of knowledge about mental disorders and their desire to learn how to manage them:*



*“There’s a big population of people who aren’t well [i.e., with mental disorders], but I don’t think they’re all the same.”*



*“As a pharmacist, I don’t necessarily have a global vision of the person’s pathology when I receive a prescription.”*



*“I admit that I’m really interested. I know a bit about psychiatry, but in private practice, you’re a little afraid of their behaviours, precisely because of a lack of knowledge.”*
Primary care provider

Psychiatric health professionals identified the need to train primary care providers to receive PMSI and to organise FG on primary care providers’ representations about mental disorders with a view to eliminating bias:


*“First, we need to train general practitioners in psychiatry so they can provide better follow-up to our patients”*
Psychiatric professional

Psychiatric health professionals also reported needing training for themselves, especially concerning the implementation of good practices and protocols for certain medical acts (e.g., obligation to prescribe check-ups, blood and urine analyses to check the biological impact of a treatment).


*“It would be perfect to (…) you prescribe a certain medicine, and you have a dedicated protocol.”*
Psychiatric professional

#### 3.2.4. Theme 4—Support

Our study revealed that all participants needed more support in their respective roles. Patients reported that they wished to attend group-based sports activities to increase their motivation:


*“Yes, going to several [group-based activities] is motivating. I’m alone, at home. I’m reluctant to go walking because I’m alone. If I had support, maybe I would feel more motivated.”*
User

Sharing experiences with peers was also described as essential to achieve better physical and mental health:


*“(…) we each have a different disease. But we live very well with it. Every once in a while, you need to talk about it, sometimes just briefly, just five minutes, because it brings you back into line a little.”*
User

Carers described themselves as exhausted and strongly expressed their desire for more comprehensive support, such as the setting-up of day hospitals, and living spaces adapted to the needs and expectations of PMSI.


*“When you get older—I mean, I think that I’m getting quite a bit older—I think you’d like someone to take over because you’re not here forever; you get older, you get tired and… You’d like to be helped more. We shouldn’t leave parents or carers like that.”*
Carer


*“They would need stimulants and full-time educators, because, right, we trained ourselves, but we’re also emotional; we’re tired because we’re all alone; we’re isolated. When there’s a crisis to manage we’re all alone, there’s no help at time t. [I dream of] an ideal place, where he [her son] would be stimulated right from the start of the day with a good breakfast, and then we could go for a walk…”*
Carer

Primary care providers and psychiatric professionals expressed that support was required for logistical organisation of patients’ medical appointments (e.g., booking, transport). Accompanying PMSI to their psychiatric care consultations (e.g., with the help of peer-health workers) was perceived as one good way to facilitate access to care:


*“(…) so they come, either with a nurse or with an educator or with a peer-health workers. That’s kind of gone now, but we used to have that, and it allowed us to at least be able to say ‘you need to go to the cardiologist’, and the peer-health worker would have helped to bring them. Like the educators can do to help them come to our office.”*
Primary care provider


*“Above all, they need to be with someone, they’re so alone all the time; the times when they’re scared, [the times when] they go to have a medical exam or [when] they’re going to be given results that they might not be able to understand…”*
Psychiatric professional

Furthermore, the development of global health computer and smartphone-based applications, as well as financial remuneration for carers for the time invested in the complex follow-up of PMSI, were also seen as facilitators to improve the quality of care received.

## 4. Discussion

The qualitative exploratory study described here was the first step of mixed-methods project COPsyCAT, whose objective is to construct and test a new cardiovascular risk prevention programme for PMSI. The study’s results provide a clear and comprehensive view of the needs of the four key populations studied here in terms of improving the cardiovascular health of PSMI. To our knowledge, it is the first study to simultaneously explore the point of view of several major types of stakeholders in the field of mental health in France.

One of the study’s most salient findings is the high level of communication difficulties between primary care outpatients and hospital psychiatric professionals. This finding is not new, and articles dating back to 1985 [30] relate similar problems. For example, a study published in 2014 [31], which aimed to better understand the practices of general practitioners in relation to mental disorders and their expectations of hospital psychiatric practitioners, highlighted that the main obstacles to efficient cooperation between general practice and hospital psychiatry services were the difficulties in referring a patient to a specialist and the stigmatisation and reluctance of patients to see one. Optimising collaboration between general practitioners and psychiatric professionals requires improved communication and training.

The effective implementation of the Shared Medical Record (*Dossier medical partagé*), a computerized patient record project initiated in 2011 by the French ministry of health [32] which was integrated into a new online governmental service called My Health Space (*Mon espace Santé*) in 2021, could help to solve these communication problems between non-hospital and hospital-based health sectors. The Shared Medical Record is one component of electronic medical records. It allows patients to communicate with healthcare providers, renew a prescription, schedule medical appointments, and view elements of their medical data, including laboratory test results. The use of this tool has shown good results in specific populations. For example, in people living with HIV, the Shared Medical Record, especially when used frequently, has been associated with maintaining high adherence, while not using it has been associated with reduced adherence over time in patients with access to it [33].

Our study also revealed several information-based needs which were not being met. Most of the information participants mentioned in the FG already exists; the challenge is to centralize these data. One of the challenges for the new cardiovascular risk prevention programme will be to bring together all the information relevant for PSMI in pleasing, user-friendly and accessible media (brochures, internet site, etc.)

Various training needs were also expressed by the study participants. Several studies have already underlined the need to introduce a training module on mental health in the core curriculum of medical studies for general practitioners [34]. The use of techniques, such as simulation, could also improve the skills of general practitioners in the management of PMSI. For example, a recent review concluded that simulation is effective for training professionals (i.e., other than psychiatrists) in psychiatry [35].

With regard to training needs for PSMI, both psychoeducation and therapeutic patient education can be effectively included in treatments [36]. They provide real benefits by helping patients play an active role in their own health care, resulting in improved treatment outcomes and quality of life [37]. The challenge when using these methods is to balance their collective nature with the individual needs of PSMI.

The cardiovascular risk prevention programme to be piloted will have to find ways to enable participants to develop skills to manage their physical health, taking into account their desires, expectations and level of knowledge of cardiovascular risks. Furthermore, carers are acknowledged as extremely important in the development of high-quality mental-health services. They play a crucial role in addressing poor physical health in PSMI [24]. Consequently, we will take their views, experiences and opinions into consideration when developing the prevention programme.

Implementing the Open Dialogue therapeutic approach in mental health services could also address some of the important needs identified in the work presented here, such as communication and greater consideration of carers’ and psychiatric service users’ points of view [38]. Open Dialogue is also a way of organizing services. It is based on a philosophy of teamwork, a democratic sharing of skills among all care professionals [39], including psychiatrists, psychologists, nurses and social workers. It consists in crisis management that is patient-centred and does not systematically resort to hospitalisation or pharmacotherapy. It focuses on the emotions experienced by the person during a crisis rather than on their symptoms. Finally, it involves integrating patients into the community.

Our study highlights the need for more support to be provided to PMSI. A peer-health worker system for PMSI and a professional peer-health worker training course were established in France in 2011. The peer-health worker is a professional whose job is to help patients by advising them on how to access their social rights and by providing relational support. The course aims to give greater legitimacy to the experiential knowledge of people who use mental health services, with a view to fostering greater consideration of PSMI in care practices [40]. Peer-health worker tasks include (i) participating in the design and implementation of an individualized care plan, in collaboration with the PMSI and an interdisciplinary healthcare team and (ii) contributing to improve the quality of care and reception PMSI receive in mental healthcare services. There are currently hundreds of professional PMSI peer-health workers throughout France [41]. The development of training and the involvement of several peer-health workers in all mental health services could help meet the need for support expressed by the different stakeholders in our study.

Finally, the difficulties associated with pharmacological treatments came up several times in participants’ discourses. These difficulties are well known in the field of psychiatry [42]. It is therefore essential we open up discussion on treatments when developing the planned cardiovascular risk prevention programme and ensure that they can be modified and interrupted, if necessary, through exchanges between health professionals, carers and PSMI.

### Limitations and Strengths

The main strength of our work is that we conducted a rigorous qualitative study with four key populations in the field of mental health. Moreover, the researchers first conducted independent coding work, which guaranteed a high degree of robustness of the results.

One of the study’s limitations is that the investigator at each participating site knew the service users and carers prior to their inclusion in the study. This may have led to selection bias. However, the fact that we had four participating sites, and four different recruiting investigators helped limit any such bias. Another limitation is that it was not possible to include all the health professionals who play a role in improving PSMI health. In the feasibility study, we will seek to involve more specialists such as physical and rehabilitation medicine (PRM) physicians.

## 5. Conclusions

This exploratory qualitative study provided us with concrete elements for the creation of the planned cardiovascular risk prevention programme for PSMI, which is currently in its final stages of development. Certain FG observations, such as the need to implement a module dedicated to mental health in the core curriculum of general practitioners’ medical studies, cannot be integrated into it. Nevertheless, in collaboration with PSMI, we have developed simple proposals such as centralizing all medical, social, cultural, sports and educational services at the local level. Thanks to this mapping of services, in collaboration with a peer-health worker, PSMI will be able to build a personalized recovery plan.

The programme will be presented to the study’s steering committee, which includes mental health users and carers, in the near future. We hope to publish the details of the programme and the results from its experimentation phase at a later date.

## Figures and Tables

**Figure 1 ijerph-19-06847-f001:**
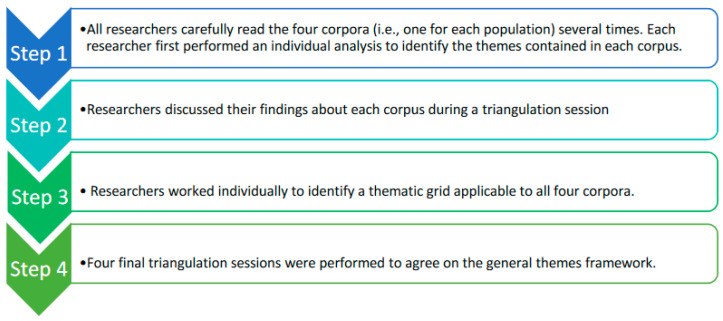
Thematic analysis process.

## Data Availability

Data and materials will be shared upon request to Marie Costa.

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
