# Peer review of "Results of a Qualitative Study Aimed at Building a Programme to Reduce Cardiovascular Risk in People with Severe Mental Illness"

_ijerph, 2022, doi:10.3390/ijerph19116847_

Round 1
Reviewer 1 Report
Please try to describe in a better way the recruitment of the probands, you need written consent and registration of the study in an official /national or international register. It would be nice if you would have randomization with a control group without intervention.
Scoring with an intelligence Test would be nice. For your purpose your need 50 patients in the interventional group.
The study is very important and I wish you good results. But your study protocol has o be improved.
Read, compare design and cite:
Exercise Improves Cognitive Function—A Randomized Trial on the Effects of Physical Activity on Cognition in Type 2 Diabetes Patients R Leischik, K Schwarz, P Bank, A Brzek, B Dworrak, M Strauss, H Litwitz, ...Journal of Personalized Medicine 11 (6), 530
Read and cite
Plasticity of Health
R Leischik, B Dworrak, M Strauss, B Przybylek, T Dworrak, D Schöne, ...
German Journal of Medicine 1 (DOI:10.19209/GJOM000001), 1-17
Author Response
Please try to describe in a better way the recruitment of the probands, you need written consent and registration of the study in an official /national or international register.
Thank you for raising this omission. We added the following information in the Ethics Approval section:
“This study is registered on the official French national clinical trials website under the number NCT03689296, and with the French National Agency for the Safety of Medicines under the number 2019-A00281-56”
We also added more detail to the recruitment section:
“Patient recruitment
The principal investigators (all physicians) working at each of COPsyCAT’s five participating sites (one investigator per site) already knew the participants. They identified PMSI who were eligible (i.e., who met inclusion criteria) for enrolment in the qualitative exploratory study, and invited them to participate in COPsyCAT during a routine follow-up visit.”
It would be nice if you would have randomization with a control group without intervention. Scoring with an intelligence Test would be nice. For your purpose your need 50 patients in the interventional group.
We agree that it would be useful to set up a randomized controlled trial exploring several different factors, including an intelligence score. The scheduled next step of the present study is to perform a feasibility study, which will involve only 30 participants. If this feasibility study is successful, we will then set up a randomised controlled trial.
The study is very important and I wish you good results. However, your study protocol has to be improved.
Read, compare design and cite:
Exercise Improves Cognitive Function—A Randomized Trial on the Effects of Physical Activity on Cognition in Type 2 Diabetes Patients R Leischik, K Schwarz, P Bank, A Brzek, B Dworrak, M Strauss, H Litwitz, ...Journal of Personalized Medicine 11 (6), 530
Thank you for sharing this article. That study’s design is very pertinent and it showed interesting results. However, its study population (i.e., Type 2 Diabetes Patients) was quite different from ours (i.e., people living with severe mental illness). Furthermore, our study was qualitative in nature, with the aim of building a health promotion programme; it did not assess the effect of physical activity.
Read and cite
Plasticity of Health, R Leischik, B Dworrak, M Strauss, B Przybylek, T Dworrak, D Schöne, ... German Journal of Medicine 1 (DOI:10.19209/GJOM000001), 1-17
Thank you for this article. We have mentioned it in the paragraph focusing on empowerment:
“It has been observed that patient and carer empowerment (1) enables better follow-up and improved health-related outcomes, including a reduction in cardiovascular disease (…)
- Leischik R, Dworrak B, Strauss M, Przybylek B, Schöne D, Horlitz M, et al. Plasticity of Health. German Journal of Medicine. 10 avr 2016;1:1‑17.
Reviewer 2 Report
Dear Editor,
Thank you for the opportunity to review this qualitative exploratory study aimed to explore the views of four populations as part of the multi-phase Copsycat project, whose objective is to build and test a cardiovascular risk prevention programme for people with severe mental illness.
The overall structure of the present manuscript is excellent however has several issues that need the author's attention.
The background argues well the topic covered, however, neglected to highlight the “advanced role” of physical and rehabilitation medicine (PRM) physicians in various interventions as well as in programs designed to reduce the cardiovascular risk in people with severe mental illness. Please add.
RESULTS AND DISCUSSION
The results are written in a very fluid and readable way for the readers.
The discussion correctly argues the results identified.
CONCLUSIONS
The conclusions are compared to the results obtained.
REFERENCES
Given the nature of this review, the following publication is need to be included:
Advanced Role and Field of Competence of the Physical and Rehabilitation Medicine Specialist in Contemporary Cardiac Rehabilitation. Hellenic J Cardiol. 2016 Jan-Feb;57(1):16-22. doi: 10.1016/s1109-9666(16)30013 6.https://pubmed.ncbi.nlm.nih.gov/26856196/
Author Response
Thank you for reviewing our manuscript and for the comments and suggestions made. We have added to the background section, and improved the strengths and limitations section, citing the suggested article as follows:
“Psychiatric health professionals and primary care providers are also essential stakeholders in caring for PSMI, and are strategically placed to promote and implement an empowerment-based approach to care. Physical and rehabilitation medicine (PRM) physicians also play an important role in interventions and programs designed to reduce cardiovascular risk (2), and consequently make an important contribution to improving cardiovascular health in PSMI.”
“Another limitation of this work is that it was not possible to include all the types of health professionals who play a role in improving PSMI health. In the feasibility study, we will try to involve more specialists, including PRM physicians”
- Papathanasiou J, Troev T, Ferreira AS, Tsekoura D, Elkova H, Kyriopoulos E, et al. Advanced Role and Field of Competence of the Physical and Rehabilitation Medicine Specialist in Contemporary Cardiac Rehabilitation. Hellenic J Cardiol. févr 2016;57(1):16‑22.
RESULTS AND DISCUSSION
The results are written in a very fluid and readable way for the readers.
The discussion correctly argues the results identified.
CONCLUSIONS
The conclusions are compared to the results obtained.
REFERENCES
Given the nature of this review, the following publication is need to be included:
Advanced Role and Field of Competence of the Physical and Rehabilitation Medicine Specialist in Contemporary Cardiac Rehabilitation. Hellenic J Cardiol. 2016 Jan-Feb;57(1):16-22. doi: 10.1016/s1109-9666(16)30013 6.https://pubmed.ncbi.nlm.nih.gov/26856196/
Thank you for this reference, which we have cited.
Reviewer 3 Report
My recommendation is to accept and publish this paper. Style and language are correct, clear and scientific.
Qualitative research are not so often represented in Journals and I think it is worth of publishing this paper, and the topic will be of interest for wider professional audience.
Author Response
Thank you for reviewing our article and for these encouraging comments.